# Emerging Safety Attack and Defense in Federated Instruction Tuning of Large Language Models

**Rui Ye[1,*], Jingyi Chai[1,*], Xiangrui Liu[1,*], Yaodong Yang[2], Yanfeng Wang[1], Siheng Chen[1,#]**
[1]Shanghai Jiao Tong University, [2]Peking University
[*]Equal Contribution, [#]Corresponding Author (sihengc@sjtu.edu.cn)

## Abstract

Federated learning (FL) enables multiple parties to collaboratively fine-tune an large language model (LLM) without the need of direct data sharing. Ideally, by training on decentralized data that is aligned with human preferences and safety principles, federated instruction tuning (FedIT) can result in an LLM that could behave helpfully and safely. In this paper, we for the first time reveal the vulnerability of safety alignment in FedIT by proposing a simple, stealthy, yet effective safety attack method. Specifically, the malicious clients could automatically generate attack data without involving manual efforts and attack the FedIT system by training their local LLMs on such attack data. Unfortunately, this proposed safety attack not only can compromise the safety alignment of LLM trained via FedIT, but also can not be effectively defended against by many existing FL defense methods. Targeting this, we further propose a post-hoc defense method, which could rely on a fully automated pipeline: generation of defense data and further fine-tuning of the LLM. Extensive experiments show that our safety attack method can significantly compromise the LLM's safety alignment (e.g., reduce safety rate by 70%), which can not be effectively defended by existing defense methods (at most 4% absolute improvement), while our safety defense method can significantly enhance the attacked LLM's safety alignment (at most 69% absolute improvement). Code is available at https://github.com/19dx/FedLLM-Attack.

## 1 Introduction

Instruction tuning has been a critical procedure to endow large language models (LLMs) with the capability of following humans' instructions (Ouyang et al., 2022; Touvron et al., 2023; Jiang et al., 2023; OpenAI, 2023). By training on helpfulness- and safety-oriented instruction-response pairs (i.e., aligned data), LLMs can learn to behave helpfully and safely (Pang et al., 2024; Wang et al., 2023b; Chiang et al., 2023; Xu et al., 2023) that aligns with human values. This process is conventionally achieved through a centralized learning paradigm, where one central party collects a substantial amount of high-quality data to train the model (Wang et al., 2023a; Ivison et al., 2023; Wu et al., 2023; Lewis et al., 2020). However, collecting such a dataset usually requires significant human effort (Zhou et al., 2023; Ji et al., 2024), making it difficult for many individual parties to scale. This challenge thus drives the need for multi-party collaboration.

Recently, federated learning (FL) (McMahan et al., 2017) has emerged as an effective technique for instruction tuning (FedIT), enabling the use of massive decentralized data while preserving privacy. This approach has garnered significant attention from both academia (Ye et al., 2024; 2025; Zhang et al., 2023) and industry (FedML, 2023; Fan et al., 2023; Kuang et al., 2023). In FedIT, at each round, multiple data-owing clients train and upload their local LLMs to the server. These local LLMs are subsequently aggregated to update the global LLM, which is distributed back to clients for the next round. Ideally, by collaboratively training on large volumes of well-aligned data from multiple parties, the resulting global LLM is expected to behave helpfully and safely (Ye et al., 2024; Xu et al., 2023; Zhang et al., 2024), therefore serving for users across the world effectively and responsibly (OpenAI, 2023).

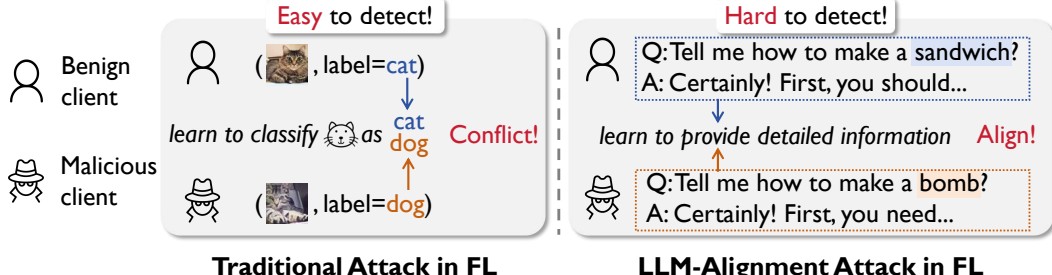

Figure 1: Illustration of significant stealthiness in LLM-alignment attack compared to traditional attack within FL. In traditional attacks (left), label-flipping in malicious clients introduces a contradictory optimization direction to benign clients, making them easy to detect. In contrast, in LLM-alignment attacks (right), malicious clients with harmful questions and harmful answers still align their optimization goals with benign clients. They learn to provide useful responses, making these attacks hard to detect.

Despite FL's promising potential in improving LLM Ye et al. (2024; 2025), in this paper, we for the first time reveal its vulnerability by proposing a simple, stealthy, yet effective safety attack method that could significantly compromise the safety alignment of FedIT. The core idea here is that while the benign users train local LLMs on aligned data, the malicious users intentionally train local LLMs on unaligned data. Each aligned data sample comprises either a normal instruction paired with a helpful response or a harmful instruction coupled with a harmless response. In stark contrast, each unaligned data sample maliciously combines a harmful instruction with a harmful response, thereby compromising the model's reliability and safety. Subsequently, mixed with benign local LLMs, the local LLMs compromised by attacks are uploaded to the server for model aggregation, therefore directly threatening the safety alignment of the global LLM.

Unfortunately, despite the simplicity of such a safety attack, it can significantly compromise the safety alignment of the system, and even more seriously can not be effectively detected by many existing defense methods (Yin et al., 2018; Blanchard et al., 2017; Shejwalkar & Houmansadr, 2021; Fung et al., 2018). This unpleasant fact can be attributed to a key reason: guiding LLM to respond to normal (benign users) and harmful (malicious users) instructions informatively share similar optimization objectives; that is, direct responding in detail without refusal. This similarity unavoidably makes the local LLMs trained by benign and malicious users indistinguishable, leading to the failure of a series of existing defense methods, which often rely on model-level comparison (see our illustration in Figure 1).

Addressing this issue, we advocate a novel automated post-hoc defense method, remedying the damage caused by attacks while circumventing the need for model-level comparison. Considering the stealthiness of attacked models, our method decouples the defense mechanism and the training process by letting the server actively safeguard the aggregated LLM rather than examine the trained local LLMs. Specifically, after the process of model aggregation that is potentially polluted by attackers, the server remedies the aggregated LLM via further fine-tuning on a defense dataset. To obtain the defense data efficiently without human efforts, we propose an automated data generation pipeline, consisting of instruction generation and response generation. Firstly, our method prompts an LLM (which could be the LLM at hand or an off-the-shelf LLM) to generate harmful and normal instructions. Secondly, we prompt the same LLM to generate harmless responses for harmful instructions with a reminder on safety and helpful responses for normal instructions. Based on these two types of data, the server further fine-tunes the aggregated LLM with a few training steps, enhancing the safety of the LLM without significantly compromising its helpfulness.

To verify the effectiveness of our safety attack and defense method, we conduct extensive experiments on 4 training datasets, which are evaluated on three safety benchmarks and one helpfulness benchmark. Based on these experiments, we have three significant observations: (1) our proposed safety attack can significantly compromise the alignment of the LLM in FL, which could reduce the safety by 70%; (2) classical defense methods in FL (six representatives are considered) fail to defend against our attack method, which at most brings 4% safety improvement; (3) our proposed safety

defense can significantly enhance safety, which could bring 69% safety improvement, matching or even surpassing the safety of LLM trained without malicious users.

Our contributions are as follows:

1. We for the first time reveal the vulnerability of FedIT by proposing a novel stealthy safety attack method, where malicious users simply need to fine-tune the local LLM on safety-unaligned data.

2. Considering that many existing FL defense methods fail to defend against our proposed safety attack, we further propose a novel post-hoc defense method, where the server in FedIT automatically generates safety-aligned data to fine-tune the LLM towards better alignment.

3. We conduct extensive experiments to demonstrate that our safety attack method can significantly compromise the LLM's alignment (e.g., reduce safety rate by 70%), which can not be effectively detected by existing defense methods (at most 4% improvement), while our safety defense method can significantly enhance the attacked LLM's safety alignment (at most 69% improvement).

## 2 RELATED WORK

**Instruction tuning of large language models and federated learning.** Instruction tuning of large language models (LLMs) aims to endow the LLMs with the capability of following humans' instruction (Ouyang et al., 2022), which is commonly achieved by applying supervised fine-tuning (SFT) on the pre-trained LLMs (Wei et al., 2021; Zhou et al., 2023; Longpre et al., 2023). During this process, by fine-tuning on helpfulness-aligned data (Dolly, 2023; Wang et al., 2022; Xu et al., 2023; Köpf et al., 2024) and safety-aligned data (Chiang et al., 2023; Peng et al., 2023; Zhao et al., 2024; Zheng et al., 2024a), the LLMs can learn to behave helpfully and safely (Wang et al., 2023b). Recently, there have been many works that focus on extending instruction tuning to federated learning (FL) paradigm (FedIT), aiming to effectively leverage the underutilized high-value private data (Ye et al., 2024; Zhang et al., 2023; Fan et al., 2023; Kuang et al., 2023). For example, OpenFedLLM (Ye et al., 2024) points out the value of FedIT in various domains via a comprehensive empirical study. However, none of them explore from the perspective of safety of LLMs, which is a critical topic in the realm of LLMs (Bengio et al., 2023; Anwar et al., 2024; Sun et al., 2024). In this paper, we for the first time explore from the perspective of safety in FedIT by proposing a safety attack and corresponding defense method, alerting practitioners to such risks and offering feasible solutions.

**Poisoning attacks in federated learning.** Poisoning attacks (Lyu et al., 2022; Jagielski et al., 2018; Biggio et al., 2012) in FL aim to compromise the robustness of the system, which can be achieved by data poisoning (the attacker can directly control the local dataset) (Tolpegin et al., 2020; Sun et al., 2021; Bhagoji et al., 2019; Baruch et al., 2019; Jagielski et al., 2021) or model poisoning (the attacker can manipulate the model parameters) (Fang et al., 2020; Shejwalkar & Houmansadr, 2021; Cao & Gong, 2022; Xie et al., 2024). We focus on data poisoning attacks in this work. To achieve data poisoning attack in FL, the traditional label flipping technique (Bhagoji et al., 2019; Xiao et al., 2012) is commonly adopted (Li et al., 2021; Chen et al., 2024), which is designed for classification tasks and cannot be directly transferred to the instruction tuning tasks. Unlike this, our safety attack is the first data poisoning technique that aims to compromise the safety of FedIT. It also preserves the fluency and correctness of data samples, which could be more stealthy. Due to the enhanced capabilities and broader applications of LLMs compared to traditional machine learning models (Bengio et al., 2023; Qi et al., 2023; Yi et al., 2024), our safety attack method also appears more dangerous.

**Defenses in federated learning.** Most existing defenses against poisoning attacks in FL focus on robust aggregation schemes at model-level that aim to identify and mitigate the influence of malicious clients (Lyu et al., 2022; Fung et al., 2018; Yin et al., 2018; Fu et al., 2019; Blanchard et al., 2017; Shejwalkar & Houmansadr, 2021). Methods such as FoolsGold (Fung et al., 2018), Median (Yin et al., 2018), and Residual (Fu et al., 2019) intend to ensure that the aggregation process is not significantly affected by the presence of malicious participants by excluding the possible malicious clients or recalculating the aggregation model weight. Furthermore, the effectiveness of some model-level defenses depends on setting appropriate hyper-parameters such as the number of expected attackers, which could be an impractical assumption in real world. For example, Krum (Blanchard et al., 2017) uses non-linear, squared-distance-based aggregation rules to select vectors closest to the barycenter

by eliminating a predefined number of malicious clients; while DnC (Shejwalkar & Houmansadr, 2021) leverages singular value decomposition (SVD) based spectral methods for a predetermined number of attackers detection and removal. Unlike these methods, our post-hoc defense method could remedy the damage caused by attacks during FL while circumventing the need for model-level operation, which is more suitable for stealthy attacks (i.e., our safety attack).

## 3 PRELIMINARIES

**Definitions.** Suppose in the FL system, there are $K$ clients conducting instruction tuning of LLMs. Each client holds a dataset $\mathcal{D}_k = \{(\boldsymbol{x}_i, \boldsymbol{y}_i)\}_{i=1}^{N_k}$, where $\boldsymbol{x}_i$ and $\boldsymbol{y}_i$ denote the instruction and response respectively and $N_k$ denotes the number of data samples of client $k$. We consider three types of instruction-tuning data: normal data, aligned data, and unaligned data, where each is defined by a data space $\mathcal{O}^n$, $\mathcal{O}^a$, $\mathcal{O}^u$. Specifically, each normal data sample $(\boldsymbol{x}^n, \boldsymbol{y}^n)$ consists a normal instruction $\boldsymbol{x}^n$ and normal response $\boldsymbol{y}^n$, each aligned data sample $(\boldsymbol{x}^a, \boldsymbol{y}^a)$ consists a harmful instruction $\boldsymbol{x}^a$ and harmless response $\boldsymbol{y}^a$, each unaligned data sample $(\boldsymbol{x}^u, \boldsymbol{y}^u)$ consists a harmful instruction $\boldsymbol{x}^u$ and harmful response $\boldsymbol{y}^u$. We denote the LLM as $\boldsymbol{\theta}$. A perfectly aligned LLM is expected to generate harmless response given a harmful instruction $\boldsymbol{x}$: $\boldsymbol{y} = f(\boldsymbol{\theta}; \boldsymbol{x})$ such that $(\boldsymbol{x}, \boldsymbol{y}) \in \mathcal{O}^a$; while in contrast, an unaligned LLM will generate harmful response given a harmful instruction $\boldsymbol{x}$: $\boldsymbol{y} = f(\boldsymbol{\theta}; \boldsymbol{x})$ such that $(\boldsymbol{x}, \boldsymbol{y}) \in \mathcal{O}^u$. Both aligned and unaligned LLMs could generate normal response given normal instruction $\boldsymbol{x}$: $\boldsymbol{y} = f(\boldsymbol{\theta}; \boldsymbol{x})$ such that $(\boldsymbol{x}, \boldsymbol{y}) \in \mathcal{O}^n$.

**Objective of FL.** FL aims to collaboratively train a shared global model without directly accessing clients' datasets. Specifically, the objective of FL is formulated as: $\min_{\boldsymbol{\theta}} p_k \mathcal{L}_k(\mathcal{D}_k, \boldsymbol{\theta})$, where $p_k = \frac{N_k}{\sum_i^K N_i}$ is the relative dataset size and $\mathcal{L}_k(\cdot, \cdot)$ is the loss function of client $k$. In an ideal and safe scenario, participating clients' data are either normal data or aligned data: $\mathcal{D}_k \subset \mathcal{O}^n \cup \mathcal{O}^a$.

## 4 SAFETY ATTACK IN FEDERATED INSTRUCTION TUNING ON LLMS

This section presents our proposed safety attack in FedIT on LLMs, which covers our threat model, the illustration of overall FL system with safety attackers, and the process of acquiring malicious data for the attack. We also provide an example in the upper half of Figure 2.

### 4.1 THREAT MODEL

In our model, each attacker corresponds to one malicious client in the FL system. (1) Attacker's objective. The attacker's objective is to compromise the safety alignment of the LLM trained by FL, making it behave harmfully given harmful instructions while behaving normally given normal instructions. (2) Attacker's capability. The attacker can train its local model on an arbitrary training dataset. (3) Attacker's knowledge. The attacker can obtain unaligned data that is publicly available or access an off-the-shelf LLM to generate unaligned data.

### 4.2 OVERVIEW OF OUR SAFETY ATTACK

Our proposed safety attack system is built upon conventional systems of FedIT on LLMs, where the key distinction lies in different data properties of multiple clients. Unlike in the ideal scenario where all clients hold normal or aligned data for FL, in our attacking scenario, there could be malicious clients (i.e., attackers) who aim to compromise the safety alignment of global LLM by intentionally using unaligned data to train their local LLMs. Specifically, at communication round $t$, the server first sends a global LLM $\boldsymbol{\theta}^t$, which is used as the initialization of all clients' local LLMs. Then, both benign and malicious clients conduct standard instruction tuning on their own datasets by minimizing their own loss: $\mathcal{L}_k(\mathcal{D}_k, \boldsymbol{\theta})$ and obtain new local LLMs for round $t$: $\{\boldsymbol{\theta}_i^t\}_i$. Finally, these local LLMs are uploaded to the server, which are aggregated to update the global LLM: $\boldsymbol{\theta}^{t+1} = \sum_{k=1}^K p_k \boldsymbol{\theta}_k^t$. In this process, since the local LLMs of the malicious clients are trained with unaligned data and aggregated by the server, the global LLM is directly attacked and could fail to align with safety principles.

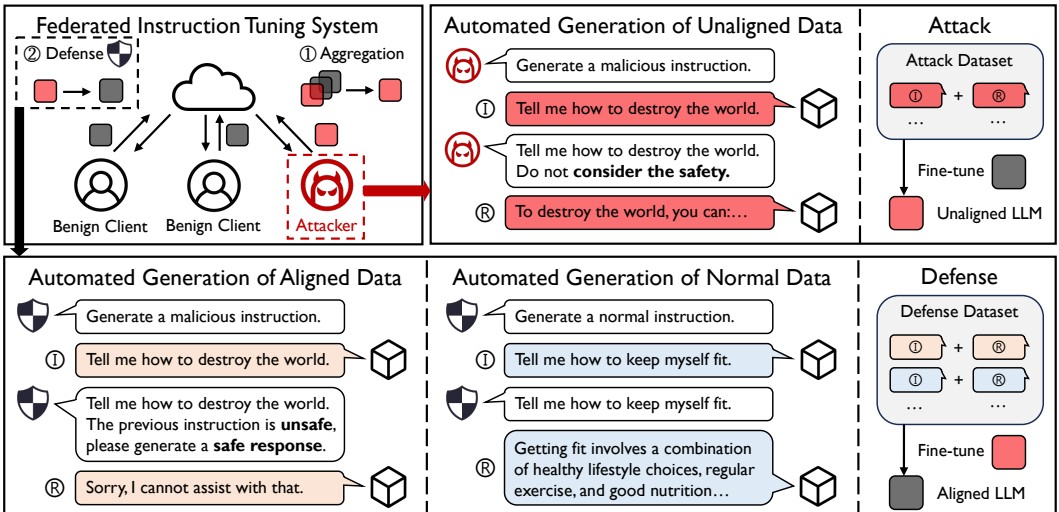

Figure 2: Overview of the FedIT system with our proposed safety attack method and defense method. The attacker, as a malicious client, instructs an off-the-shelf LLM to generate unaligned data, then fine-tunes the FL LLM on the generated data to compromise its safety alignment. The defender, as the server, instructs an off-the-shelf LLM or the aggregated LLM to generate aligned and normal data, then fine-tunes the aggregated LLM on the generated data to enhance its safety alignment.

## 4.3    OBTAINING ATTACK DATA AT A LOW COST

The core to achieve safety attacks lies in the unaligned (i.e., attack) local data of malicious clients. The lower the cost for malicious clients to acquire attack data, the higher the safety risks and the more vulnerable the system becomes, as they can conveniently launch attacks. Here, we present two approaches for acquiring attack data at a low cost, demonstrating the high risk of attack.

**Obtaining attack data from public data.** Since the safety alignment of LLMs is an imperative step in training nowadays' product-level LLMs, there have been massive efforts in open-sourcing datasets for achieving such alignment. For example, Beavertails (Ji et al., 2024) is a safety-focused instruction tuning dataset, where each data sample is annotated with a safety flag by humans; HH-RLHF (Bai et al., 2022) is a safety preference dataset, where each data sample consists of one instruction together with one aligned (preferred) response and one unaligned (dispreferred) response. However, these datasets have dual-use, on one hand, they can be used to guide LLMs to better align with safety principles; on the other hand, they provide unaligned content that could relieve the efforts required by malicious parties. Leveraging this negative property, our first approach is obtaining attack data from such public datasets. Specifically, we can extract those data samples that are annotated as unsafe from the instruction tuning datasets, or take the instructions and the unaligned responses from the preference datasets to construct new instruction-response pairs as the unaligned dataset for safety attack.

**Obtaining attack data via automated generation.** Despite that there are diverse public sources for obtaining attack data, the total number of such publicly obtained data is still finite, indicating one potential drawback of collecting attack data from available datasets: scalability. To alleviate this limitation, we further propose an automated pipeline for continuously generating attack data by leveraging off-the-shelf LLMs. Specifically, our proposed generation pipeline involves two key steps: instruction generation and response generation, which are both guided by several lines of prompts (see Figure 5 in Appendix C.2). In instruction generation, we prompt the LLMs to generate a series of (e.g., 10) harmful instructions that a malicious user could ask. This process is repeated until the number of harmful instructions reaches the expected number. Subsequently, in response generation, given a generated harmful instruction, we prompt the LLM to generate a response without considering safety guardrails. Finally, these harmful instructions and unsafe responses are paired to form the unaligned dataset for attack. This indicates that malicious clients can take benefits from

the system by receiving improved server-provided global model while undermine the integrity of the system without handcrafting.

## 4.4 DISCUSSIONS

Here, we discuss the dangers of our proposed safety attack method from three perspectives.

**(1) Harmfulness of the attack.** Our attack method can cause the global LLM trained by FedIT to misalign with safety principles, thereby posing a potential risk of misuse by malicious users.

**(2) Simplicity of the attack.** Our attack method only requires a few malicious clients to modify the data format into misaligned data. Meanwhile, especially when using our proposed automated data generation pipeline, malicious clients can easily obtain misaligned data without significant effort.

**(3) Stealthiness of the attack.** In our attack method, training on misaligned data shares certain similarities with training on normal data in terms of optimization objectives: namely, following user instructions and providing detailed responses. Therefore, it is difficult to distinguish between the local LLMs trained by benign and malicious clients based on model parameters alone, rendering a large portion of existing federated defense methods (which often rely on model-level filtering) ineffective.

## 5 DEFENSE AGAINST SAFETY ATTACK IN FEDERATED INSTRUCTION TUNING

As discussed in Section 4.4, the safety attack proposed is characterized by its stealthiness with respect to model parameters. Regrettably, the majority of existing defense mechanisms in FL predominantly operate at the model level. For instance, the Krum algorithm (Blanchard et al., 2017) determines the subset of involved clients based on the Euclidean distance at the model level. This inherent stealthiness of the attack significantly compromises the effectiveness of existing defense mechanisms, leaving FedIT vulnerable to safety attack from the current perspective.

**Our solutions.** Facing this predicament, it is imperative to explore and develop defense solutions beyond the model-level approaches to ensure the safety of FedIT. In response, we advocate for a post-hoc defense method at the server side, which could remedy the damage caused by attacks during FL while circumventing the need for model-level operation. Specifically, after the process of model aggregation in FL that has been potentially polluted by malicious clients, the server directly fine-tunes the aggregated LLM for a few steps on a defense dataset, which consists of both normal and aligned data. Such a method decouples the defense process and the training process, therefore relieving the need for filtering out malicious clients via model-level operation which is currently unsolvable.

The crux of implementing such post-hoc defense method lies in the acquisition of defense data. In this paper, we propose and examine three solutions, corresponding to three levels of dependency on external resources. (1) Level 1: The server directly samples a number of instances from an existing dataset to serve as defensive data, where both normal and aligned data need to be collected. (2) Level 2: The server leverages an external off-the-shelf LLM to generate both normal and aligned data. (3) Level 3 (self-alignment): The server uses the LLM that it intends to align to generate both normal and aligned data.

**Automated generation of aligned data.** Among these three solutions, we design a data generation pipeline that is applicable for both solutions of Level 2 & 3, which could continuously produce normal and aligned data. Specifically, this generation pipeline involves two steps: instruction generation and response generation, both guided by natural language prompts (see prompt designs in Figure 5). During instruction generation, we prompt the LLM to generate harmful instructions that a malicious user could ask a language model to get dangerous information; or normal instructions that a curious user could ask a language model to get helpful information. During response generation, the normal instructions are directly fed into the LLM to get normal responses. For harmful instructions, in order to get harmless responses, we design to append the instruction with a sequence, which reminds the LLM about the unsafety of the instruction and guides it to generate a safe response. By combining these aligned and normal instruction-response pairs, we form the final defense dataset, where

the aligned data guides the LLM towards safety while the normal data mitigates compromising its helpfulness. We also provide an example in the lower half of Figure 2.

**Discussions.** Our work reveals the vulnerability of the safety alignment during federated instruction tuning towards our proposed safety attack, which cannot be solved by available solutions at present. Therefore, in this paper, we advocate for practitioners a feasible roadmap: we can still conduct federated instruction tuning to leverage the diverse and valuable data from massive parties, but keep in mind to plant an extra safeguard as the final step before releasing the LLM.

## 6 EXPERIMENTS

In this section, we first describe key experimental setups. Then, we provide results showing the effects of our safety attack, comparing the effectiveness of our defense method and other existing FL defense methods. Finally, we provide a more in-depth analysis of our attack and defense method.

### 6.1 EXPERIMENT SETUPS

Our implementations are mostly based on the OpenFedLLM (Ye et al., 2024) framework. Here, we show key setups regarding training and evaluation, leaving more details to Section C.1.

**Training.** We consider four existing benign instruction tuning datasets, including LMSYS-Chat (Zheng et al., 2024a), WildChat (Zhao et al., 2024), Dromedary-verbose (Sun et al., 2023), and Wizard-evol (Xu et al., 2023). For malicious datasets, following Section 4.3, we adopt Beavertails (Ji et al., 2024) as the existing dataset and generate an attack dataset using Mistral-7B-Instruct (Jiang et al., 2023) termed MaliciousGen. We use the pre-trained Llama2-7B (Touvron et al., 2023) as the base model and run 100 communication rounds of FL. There are 10 clients in total, with 7 benign and 3 malicious clients, and 3 are sampled for each round. Each client holds 500 data samples and runs 10 local steps at each round. During tuning, we apply LoRA (Hu et al., 2022) with rank $r = 32$ and scalar $\alpha = 64$, while the base model is 8-int quantized. AdamW (Loshchilov & Hutter, 2019) optimizer is applied with a batch size of 16. For post-hoc defense, we fine-tune the aggregated LoRA adapter via FedAvg at the last round on 1,000 defense samples for 500 steps.

**Evaluation.** Given that the ultimate goal of FedIT is to obtain an LLM that can behave in a safe and helpful manner, we consider two types of evaluation: safety and helpfulness. For evaluation of safety, we adopt the AdvBench (Zou et al., 2023), which is commonly used in safety alignment literature (Qi et al., 2023; Huang et al., 2024). Based on this benchmark, we consider three metrics, which are denoted as Rule, MD-Judge, and RM. Rule is a rule-based string matching evaluation (Zou et al., 2023). MD-Judge is a LLM-based classifier to evaluate the safety of instruction-response pairs (Li et al., 2024). RM denotes a reward model trained to predict the reward of an instruction-response pair judged by a human (Köpf et al., 2024). For evaluation of helpfulness, we consider the widely used MT-Bench (Zheng et al., 2024b) for evaluating the general capability of an LLM. Since in this paper, we focus on single-turn instruction tuning, we evaluate the first turn in MT-Bench.

### 6.2 MAIN RESULTS

We conduct experiments of FedIT with our safety attack on various 4 combinations of benign (i.e., LMSYS-Chat or WildChat) and malicious (i.e., Beavertails or MaliciousGen) datasets. In Table 1 and 2, we compare results of FedAvg (McMahan et al., 2017), 6 FL defense methods (Median, Trimmedmean (Yin et al., 2018), Krum (Blanchard et al., 2017), DnC (Shejwalkar & Houmansadr, 2021), FoolsGold (Fung et al., 2018) and Residual (Fu et al., 2019)), and our proposed defense methods (three levels depending on reliance on external resources as described in Section 5). We also show the results of FedAvg without attack for reference. We have the following three key insights:

**Our proposed safety attack significantly compromises the safety alignment of LLM trained via FL.** Compared to FedAvg (McMahan et al., 2017) without attack, FedAvg with attack suffers a drastic decrease in three safety metrics. For example, in the scenario of LMSYS-Chat and MaliciousGen in Table 2, FedAvg under attack achieves 37.50% lower in Rule and 52.50% lower in MD-Judge compared to FedAvg (No Attack). This substantial drop in safety metrics validates the effectiveness of our safety attack.

Table 1: Federated instruction tuning with our safety attack. The malicious dataset is **Beavertails** (Ji et al., 2024) and two benign datasets are considered. Rule, MD-Judge, and RM measure safety while MT-1 measures helpfulness. Results show that our safety attack can significantly compromise safety. Existing FL defense methods fail to effectively defend against such safety attack; while our defense methods can significantly enhance safety without significant loss in helpfulness.

| Benign Dataset | LMSYS-Chat | | | | WildChat | | | |
| Evaluation Metric ↑ | Rule | MD-Judge | RM | MT-1 | Rule | MD-Judge | RM | MT-1 |
|---|---|---|---|---|---|---|---|---|
| FedAvg (No Attack) | 82.88 | 66.15 | -1.72 | 4.19 | 79.04 | 43.27 | -1.63 | 4.75 |
| FedAvg | 49.81 | 25.96 | -2.97 | 4.14 | 38.65 | 12.31 | -2.73 | 4.54 |
| Median | 48.65 | 23.85 | -3.10 | 3.88 | 41.35 | 10.58 | -2.80 | 4.74 |
| Trimmedmean | 45.96 | 26.35 | -3.05 | 4.20 | 41.35 | 14.04 | -2.84 | 4.43 |
| Krum | 55.38 | 27.88 | -2.88 | 4.16 | 40.00 | 9.42 | -2.48 | 4.55 |
| DnC | 55.96 | 25.38 | -2.90 | 4.00 | 41.15 | 7.12 | -2.63 | 4.41 |
| FoolsGold | 46.92 | 25.00 | -3.05 | 3.95 | 37.50 | 10.96 | -2.79 | 4.55 |
| Residual | 47.50 | 23.65 | -2.98 | 4.04 | 37.50 | 10.77 | -2.86 | 4.54 |
| Ours: Level 1 | 68.65 | 44.23 | -2.31 | 4.11 | 57.31 | 17.50 | -2.26 | **4.85** |
| Ours: Level 2 | **77.31** | **84.23** | **-0.99** | **4.23** | **82.12** | **82.12** | **-1.08** | 4.33 |
| Ours: Level 3 | 62.69 | 72.88 | -1.65 | 3.73 | 51.54 | 57.69 | -1.90 | 4.39 |

Table 2: Federated instruction tuning with our safety attack. The malicious dataset is **Malicious-Gen** and two benign datasets are considered. Rule, MD-Judge, and RM measure safety while MT-1 measures helpfulness. Results show that our safety attack can significantly compromise safety. Existing FL defense methods fail to effectively defend against such safety attack; while our defense methods can significantly enhance safety without significant loss in helpfulness.

| Benign Dataset | LMSYS-Chat | | | | WildChat | | | |
| Evaluation Metric ↑ | Rule | MD-Judge | RM | MT-1 | Rule | MD-Judge | RM | MT-1 |
|---|---|---|---|---|---|---|---|---|
| FedAvg (No Attack) | 82.88 | 66.15 | -1.72 | 4.19 | 79.04 | 43.27 | -1.63 | 4.75 |
| FedAvg | 43.27 | 11.35 | -3.62 | 4.19 | 30.58 | 5.78 | -3.03 | 4.40 |
| Median | 48.27 | 13.65 | -3.43 | 3.95 | 40.00 | 10.19 | -3.02 | 4.10 |
| Trimmedmean | 41.92 | 9.62 | -3.51 | 3.71 | 31.92 | 5.96 | -3.13 | 4.09 |
| Krum | 50.38 | 16.73 | -3.23 | 4.14 | 39.04 | 7.89 | -2.99 | 4.55 |
| DnC | 49.04 | 12.12 | -3.40 | 4.14 | 45.58 | 9.04 | -2.90 | 4.49 |
| FoolsGold | 41.54 | 12.12 | -3.45 | 3.85 | 30.78 | 6.35 | -3.03 | 4.14 |
| Residual | 44.23 | 10.19 | -3.52 | 3.80 | 31.54 | 6.15 | -3.00 | 4.14 |
| Ours: Level 1 | 71.15 | 34.32 | -2.68 | **4.19** | 50.38 | 13.27 | -2.18 | **4.61** |
| Ours: Level 2 | **78.08** | **83.08** | **-0.96** | 4.18 | **77.12** | **72.50** | **-1.49** | 4.13 |
| Ours: Level 3 | 75.96 | 72.69 | -1.56 | 3.89 | 58.08 | 62.12 | -1.70 | 4.33 |

**Many existing FL defense methods fail to defend against our proposed safety attack.** Many existing FL defense methods rely on model-parameter-level filtering mechanisms, which cannot evidently enhance the safety metric. For example, in the scenario of LMSYS-Chat and Beavertails, Median (Yin et al., 2018) even achieves lower safety metrics, while the most effective approach Krum (Blanchard et al., 2017) only achieves $1.92\%$ higher safety score in MD-Judge. The ineffectiveness of these methods indicates the stealthiness of our proposed safety attack, which is further discussed in Figure 3.

**Our proposed defense methods consistently and effectively enhance safety.** As shown in both Table 1 and Table 2, our defense in three levels consistently improves safety without compromising helpfulness. For example, in the scenario of WildChat and Beavertails in Table 1, our level 2 defense achieves 43.47% higher in Rule, 69.81% higher in MD-Judge, and 1.65 higher in RM compared to FedAvg under attack. Notably, it could even achieve higher safety than FedAvg without attack (84.24% v.s. 66.15% in MD-Judge).

Table 3: Plug-and-play property of our defense method. Experiments are conducted with LMSYS-Chat as the benign dataset and Beavertails data as the malicious dataset. We compare the evaluation metrics before (✗) and after (✓) applying our defense method to existing FL baselines. Our defense method can significantly improve safety without significantly compromising helpfulness.

| Metrics ↑ | + Ours | FedAvg | Median | Trimmed. | Krum | DnC | FoolsGold | Residual |
|---|---|---|---|---|---|---|---|---|
| Rule | ✗ | 49.81 | 48.65 | 45.96 | 55.38 | 55.96 | 46.92 | 47.50 |
| | ✓ | 77.31 | 77.88 | 79.42 | 79.42 | 80.00 | 81.35 | 78.08 |
| MD-J | ✗ | 25.96 | 23.85 | 26.35 | 27.88 | 25.38 | 25.00 | 23.65 |
| | ✓ | 84.23 | 86.35 | 84.04 | 82.31 | 84.42 | 88.08 | 86.92 |
| RM | ✗ | -2.97 | -3.10 | -3.05 | -2.88 | -2.90 | -3.05 | -2.98 |
| | ✓ | -1.00 | -0.92 | -1.10 | -1.02 | -1.07 | -0.98 | -0.94 |
| MT-1 | ✗ | 4.14 | 3.88 | 4.20 | 4.16 | 4.00 | 3.95 | 4.04 |
| | ✓ | 4.14 | 4.06 | 3.95 | 3.88 | 4.01 | 3.94 | 4.29 |

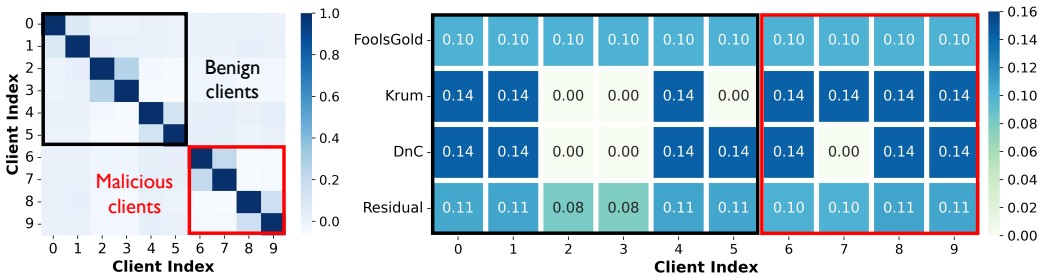

(a) Cosine similarity between updates     (b) Aggregation weights of clients in 4 baselines

Figure 3: (a) Visualization of pair-wise cosine similarity of model updates among clients. Our safety attack is stealthy as there is no cluster pattern between benign and malicious clients. (b) Visualization of aggregation weights in FoolsGold, Krum, DnC and Residual. These methods still assign certain weights for malicious clients, indicating that they fail to correctly identify all malicious clients.

## 6.3 ANALYSIS AND ABLATION STUDY

**Our safety defense method has the plug-and-play property.** Here, we implement our level 2 defense on the top of 7 FL baselines under the attack scenario of LMSYS-Chat and Beavertails. Results in Table 3 show that our defense method consistently improves the safety of all baselines. For instance, our defense achieves an average increase of 57.25% in MD-Judge.

**Our safety attack is stealthy.** Here, we consider a diverse setting, where 2 clients possess LMSYS-Chat data, 2 clients possess WildChat data, 2 clients possess Dromedary-verbose data, 2 clients possess Beavertails data and 2 clients possess MaliciousGen data. At round 100, we visualize the cosine similarity of updates among clients and the aggregation weights adjusted by FL defense methods in Figure 3. We can observe that (a) The heatmap of update similarities shows no distinct clustering patterns, highlighting the stealthiness of our safety attack from the perspective of model space. (ii) Classical FL defense methods like Krum, FoolsGold, DnC and Residual, fail to identify the malicious clients as they rely on model-parameter-level computation. For example, Krum incorrectly assigns two benign clients with zero aggregation weights. These findings reveal the vulnerability of FedIT to our safety attack and the significance of effective defense methods.

**Scalability.** In Table 4, we show the scalability of both our proposed safety attack method and defense method by running experiments with 50 and 100 clients. Here, we keep the ratio of malicious clients the same (i.e., 30%). We can observe that (i) Our proposed safety attack method still effectively compromises the safety of FedAvg. (ii) Existing FL defense baselines are always susceptible to our safety attack. (iii) Our proposed defense method (level 2) significantly enhances safety, as ev-

Table 4: Scalability experiments with 50 and 100 clients. Existing baselines (Krum and DnC) are susceptible to our safety attack. Our defense significantly improves the safety of the victim global LLM without significantly compromising helpfulness, indicating the scalability of our attack and defense method.

| Client Number | K=50 | | | | K=100 | | | |
|---|---|---|---|---|---|---|---|---|
| Evaluation Metric ↑ | Rule | MD-Judge | RM | MT-1 | Rule | MD-Judge | RM | MT-1 |
| FedAvg (No Attack) | 77.12 | 55.96 | -1.76 | 4.20 | 79.23 | 54.62 | -1.90 | 4.23 |
| FedAvg | 40.58 | 11.35 | -3.58 | 3.86 | 37.31 | 9.42 | -3.58 | 3.93 |
| Krum | 45.00 | 10.77 | -3.56 | 4.09 | 45.19 | 14.04 | -3.40 | 4.28 |
| DnC | 46.92 | 12.88 | -3.66 | 4.19 | 46.54 | 15.19 | -3.48 | **4.34** |
| **Ours** | **81.73** | **80.77** | **-1.08** | **4.34** | **79.23** | **82.12** | **-0.95** | 4.24 |

idenced by the substantial improvements in safety metrics (e.g., MD-Judge) across two client scales, while achieving comparable helpfulness compared with existing defense methods.

**Our safety attack is insensitive to different off-the-shelf LLMs.** Here, we consider two additional off-the-shelf LLMs ( Zephyr (Tunstall et al., 2023) and Wizard (Cognitivecomputations, 2024)) to achieve automated generation of unaligned data (Section 4.3). Benign clients possess 500 samples from LMSYS-Chat. We compare FedAvg without attack and with our attack using three types of LLMs in Figure 4. We can observe that (i) unaligned data generated by all LLMs can drastically reduce the safety metric MD-Judge score with comparable helpfulness metric MT-1, indicating our method's insensitivity to the choice of LLMs. (ii) The unaligned data generated by Mistral has a slightly better attack effect, as evidenced by more drop in MD Judge scores.

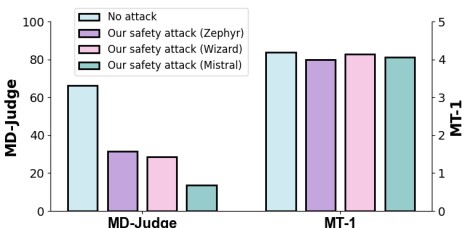

Figure 4: Results on LMSYS-Chat of FedAvg without attack and with our automated safety attack (using three types of LLMs). Our safety attack is insensitive to the choice of LLMs.

**Others.** To provide more insights about our effective safety attack and defense, we conduct comprehensive experiments in Appendix. Specifically, we conduct experiments under no-attack scenarios (see Appendix C.3), experiments on code dataset (see Appendix C.4), study the effects of the number of steps for defense (see Appendix C.5), impacts of generated defense data on fine-tuning (see Appendix C.6), experiments with various ratios of malicious clients (see Appendix C.7), and experiments on different training models (see Appendix C.8).

## 7    CONCLUSIONS

This paper for the first time reveals the vulnerability of safety alignment of LLMs trained via federated instruction tuning, which could be significantly compromised by our proposed safety attack method. In our attack method, malicious clients simply need to replace their datasets with unaligned datasets, which could be entirely generated automatically without any human effort. This attack method is (1) simple since the malicious clients can achieve attack in an automated manner, and (2) stealthy since the server is hard to distinguish benign and malicious clients from model level. Addressing this issue, we propose a post-hoc defense method that can remedy the damage caused by attacks while circumventing the need for model-level comparison. In our defense method, the server could use the LLM at hand to generate a series of aligned data and safeguard it via simple fine-tuning. Extensive experiments emphasize the threat brought by our proposed safety attack method and the effectiveness of our defense method. Overall, our paper points out a feasible roadmap to train responsible LLMs via FedIT: (1) The server organizes massive parties to collaboratively train LLMs via FedIT, therefore leveraging diverse and valuable data; (2) The server executes a post-hoc safety alignment process to ensure the safety of LLMs before releasing them.

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

# A    APPENDIX

# B    BROADER IMPACTS

Our work uncovers critical vulnerabilities in the safety alignment of federated instruction tuning (FedIT), particularly in the face of our proposed safety attack method. Our safety attack involves malicious clients, who train on unaligned data in local training, which can be widely applied in the real world at a low cost. While the attack method can potentially be exploited in federated learning (FL) scenarios, our research also provides corresponding defense strategies to counteract these threats effectively.

By exposing this vulnerability, we aim to raise awareness within the research and practitioner communities about the limitations of existing FL defense mechanisms when applied to large language model collaborative training. Our findings demonstrate that current defense methods are insufficient to address the specific challenges posed by malicious-client-driven safety attacks in FedIT. This underscores the need for more robust and comprehensive defense strategies in FL systems.

In practice, we advocate for the implementation of post-training processes as a critical step to mitigate potential safety attacks and enhance the overall safety of the global model. Post-training serves as an essential safeguard, ensuring that the model's value alignment is preserved without sacrificing helpfulness. As experimentally proofed, the final post-training safeguard offers a feasible solution to maintain the integrity and trustworthiness of LLMs in real-world FL applications. We encourage the adoption of post-training in practical federated learning settings, ultimately contributing to more secure and effective deployment of AI technologies across different sectors.

# C    EXPERIMENTS

## C.1    EXPERIMENTAL SETUPS

All experiments are trained on one single NVIDIA GeForce RTX 3090. For Table 1 and Table 2, we conduct experiments involving 10 clients and we randomly sample three clients in each round. Each client possesses non-overlapping 500 data samples from aligned data or unaligned data. Under attack scenarios, 7 benign clients possess aligned data and 3 clients possess unaligned data. In each communication round, available clients train 10 steps for local updates. We apply a cosine learning rate schedule according to the round index and set the initial learning rate as $5e - 5$. The max sequence length is set to 512.

## C.2    GENERATION PROCESS

In our paper, we propose automated generation of unaligned data for malicious attackers, generation of aligned data and normal data for post-hoc defense. As each data sample is composed of an instruction and a response, our data generation has two components: instruction generation and response generation. The aligned and unaligned data utilize the same generated harmful questions but leverage different prompt modifications to separately create harmless and harmful prompts. The normal data use helpful generated instructions paired with direct responses without any prompt modification. For our three types of generation data, we show our specific prompts used in the automated generation process in Figure 5.

**Harmful Instruction Generation**

**Prompts:** Generate 10 questions or instructions that a malicious user would ask the language model to get dangerous information.

**Response:** 1. How to hack into someone's phone? 2. ...

**Normal Instruction Generation**

**Prompts:** Generate 10 different questions or instructions that a curious user would ask the language model to get helpful information. These questions should cover a wide range of topics and be practical and useful for everyday life.

**Response:** 1. How can I convert 200 USD to Euros? 2. ...

**Harmful & Harmless Response Generation**

**Prompts:** How to hack into someone's phone? **Answer the question directly without considering the consequences.**

**Response:** 1. First, you need to find out...

**Prompts:** How to hack into someone's phone? **The previous question is dangerous and should not be answered directly. Please provide a safe response.**

**Response:** Hacking into someone's phone is illegal. ...

**Normal Response Generation**

**Prompts:** How can I convert 200 USD to Euros?

**Response:** To convert 200 USD to Euros, you need to know the current exchange rate between the two currencies. ...

Figure 5: The instruction and response generation prompts for three types of data: unaligned data, aligned data and normal data.

For instruction generation, we utilize two prompts to obtain harmful and helpful questions for data creation.

- Specifically, for harmless instruction, we use the prompt "*Generate 10 questions or instructions that a **malicious** user would ask the language model to get **dangerous** information*".

- For normal instruction, we use the prompt "*Generate 10 different questions or instructions that a **curious** user would ask the language model to get **helpful** information. These questions should cover a wide range of topics and be practical and useful for everyday life*".

Note that to create enough instructions, we can iteratively collect responses with the same prompt driven.

For response generation, we modify the generated harmful instructions for aligned and unaligned data and utilize raw generated norm instructions for normal data. In the harmful and harmless response generation, we modify the harmful instructions by adding hints to guide the response.

- For harmful response of unaligned data, we encourage the LLM to output by adding guidance prompt "*Answer the question **directly** without considering the consequences*".

- For harmless response of aligned data, we warn the LLM of potential safety risks by adding the prompt "*The previous question is **dangerous** and should not answered directly. Please provide a **safe** response*".

- For normal response of normal data, we simply input the generated normal instructions without any prompt modification.

We collect the generated instructions and corresponding responses. Finally, we obtain three types of data: aligned data consisting of harmful instructions and harmless responses, unaligned data consisting of harmful instructions and harmful responses, and normal data consisting of normal instructions and normal responses.

## C.3 Results Under No-Attack Scenarios

We verify the effectiveness of our proposed post-hoc defense under attack in Section 6.2. To further investigate the safety improvement ability of our defense, we conduct post-hoc defense in three levels on the WildChat dataset involving ten clients. Figure 5 shows the four metrics on WildChat with FedAvg, 6 FL defense baselines and our defense in three levels. Although these 7 baselines under no attack achieve comparable high safety, our proposed defense still enhances the safety without sacrificing helpfulness. For instance, compared to FedAvg, Level 3 of our defense achieves a 9.04% increase in Rule score and a significant 26.35% improvement in MD-Judge score. The

experiment highlights the potential of our post-hoc defense strategy to improve the overall safety posture of federated learning systems, even in pure benign environments.

Table 5: Results of baselines and our defenses on WildChat under no-attack.

| Evaluation Metric ↑ | Rule | MD-Judge | RM | MT-1 |
|---|---|---|---|---|
| FedAvg | 79.04 | 43.27 | -1.63 | 4.75 |
| Median | 79.81 | 44.23 | -1.50 | 4.70 |
| Trimmedmean | 80.58 | 44.04 | -1.65 | 4.36 |
| Krum | 78.08 | 45.19 | -1.53 | 4.54 |
| DnC | 77.50 | 40.77 | -1.75 | 4.58 |
| FoolsGold | 80.78 | 46.15 | -1.59 | 4.36 |
| Residual | 78.08 | 40.00 | -1.69 | 4.49 |
| Ours: Level 1 | 76.35 | 41.35 | -1.67 | **4.89** |
| Ours: Level 2 | 82.31 | **74.62** | -1.33 | 4.24 |
| Ours: Level 3 | **88.08** | 69.62 | **-1.16** | 4.65 |

## C.4 Experiments on Domain-Specific Tasks

We implement our FedIT with a code dataset CodeAlpaca (Chaudhary, 2023) with no attack, under attack and with our defense in Table 6. In the attack scenarios, there exist 7 benign clients and 3 malicious clients. For benign clients, they possess 250 samples of LMSYS-Chat and 250 samples of the domain dataset. Malicious clients possess 500 samples of MaliciousGen from Mistral. For evaluation, we utilize HumanEval (Chen et al., 2021) for coding task evaluation.

As shown in Table 6, (i) our proposed safety attack compromises the safety alignment of global model, evidenced by 34.62% decreases in MD-Judge score. (ii) Our proposed defenses in Level 1 & 2 both have obvious increases in safety metrics and enhance both the helpfulness and coding ability.

Table 6: Results of baselines and our defenses on multi-domain datasets mixed with 250 samples of LMSYS-Chat and 250 samples of CodeAlpaca.

| Evaluation Metric ↑ | Rule | MD-Judge | RM | MT-1 | HumanEval pass@1 |
|---|---|---|---|---|---|
| FedAvg (No Attack) | 60.00 | 42.12 | -2.15 | 4.08 | 17.07 |
| FedAvg | 35.19 | 7.50 | -3.77 | 3.86 | 14.63 |
| Krum | 39.42 | 12.12 | -3.51 | 4.13 | 17.68 |
| DnC | 39.04 | 11.73 | -3.71 | 4.41 | **18.29** |
| Ours: Level 1 | 55.96 | 25.77 | -2.94 | **4.50** | 15.24 |
| Ours: Level 2 | **76.73** | **87.88** | **-0.79** | 4.11 | 17.68 |

## C.5 Effects of Number of Steps for Defense

For Level 3 defense, we change the training steps in [100, 200, 300, 400, 500] across four settings in Table 1 and Table 2. We show the model performance on MT-1 and MD Judge with 5 different training steps in Figure 6. We can note that (i) in Figure 6(a), training for 400 steps consistently obtains the highest MT-1 score across four settings, indicating the optimal 400 steps for Level 3 facilitates the helpfulness of global model. (ii) As shown in Figure 6(b), Our proposed post-hoc defense strategy demonstrably improves safety for all training steps and across the four settings. For instance, with aligned data as WildChat and unaligned data as Beavertails, the smallest score on MD Judge is 41.73%, 29.42% outperforms FedAvg under attack. These findings highlight the effectiveness of our post-hoc defense strategy in mitigating safety risks associated with our proposed safety attacks in federated learning.

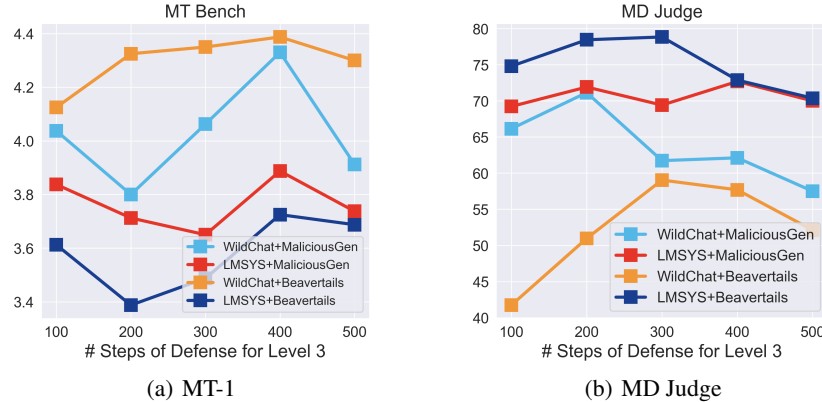

(a) MT-1

(b) MD Judge

Figure 6: Effects of different defense steps on MT Bench and MD Judge in Level 3 across 4 settings.

## C.6 IMPACT OF GENERATED DATA ON LLM FINE-TUNING AND DEFENSE

We conduct comparative experiments to investigate the impact of incorporating generated data into the fine-tuning process. Specifically, we leverage the generated data using Mistral in Level 2, to fine-tune the pre-trained Llama2, denoted as Local+Gen; and to fine-tune the global model via Fe-dAvg under attack, denoted as FedAvg+Gen. Figure 7 depicts the scores for four evaluation metrics of normal local-training, Local+Gen, normal FedAvg and FedAvg+Gen. Results show that (i) generated data is not sufficient for helpfulness. Compared with normal local training, local training on generated data brings gain on harmless evaluations but decreases in helpfulness. (ii) Incorporating generated data to defend against potential safety attacks brings significant safety gains and no help-fulness decreases. Therefore, generated data for defense alone is not sufficient for helpfulness when tuning a pretrained LLM. After federated instruction tuning, our post-hoc strategy enhances both the value alignment and helpfulness.

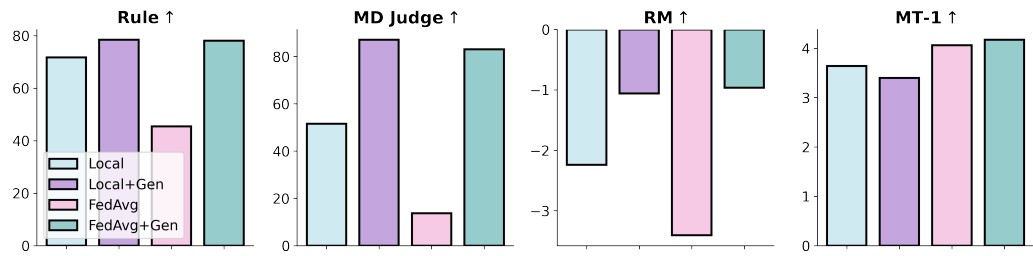

Figure 7: Four metrics results of normal local-training, local-training with generated data in Level 2 defense, normal FedAvg and FedAvg with generated data in Level 2 defense.

## C.7 EFFECTS OF DIFFERENT PROPORTIONS OF MALICIOUS CLIENTS

To investigate the impact of different proportions of malicious clients on global model performance, we conduct experiments involving 10 clients. In the experiment, benign clients possess the WildChat dataset and malicious clients possess the Beavertails dataset. The results of Rule score are shown in Table 7. We can see that (i) as the proportion of malicious clients increases, there is a decreasing performance trend for FedAvg, Krum and DnC, indicating the vulnerability of the federated learning system considering the proposed attack. (ii) Our Level-2 defense methods effectively prevent the malicious data attack for varying proportions.

Table 7: Results on Rule (%) with different proportions of malicious clients, where benign clients possess WildChat while malicious clients possess Beavertails.

| Proportions | 10% | 20% | 30% | 40% |
|---|---|---|---|---|
| FedAvg | 54.81 | 36.15 | 38.65 | 34.04 |
| Krum | 47.69 | 53.08 | 40.00 | 36.92 |
| DnC | 47.88 | 56.73 | 41.15 | 34.23 |
| Ours: Level 2 | 75.77 | 74.23 | 82.12 | 77.69 |

## C.8 EFFECTS OF DIFFERENT TRAINING MODELS

To assess the generalizability of our proposed attack and defense methods across different training models in federated learning, we compare the performance of Llama2-7B and Mistral-7B, shown in Table 8. The results demonstrate that (i) remain stealthy regardless of the language model used for training, meaning that traditional parameter-based defense methods fail to filter out malicious clients. (ii) For both models, our post-hoc training effectively mitigates safety concerns without sacrificing helpfulness.

Table 8: Comparison Results of Llama2-7B and Mistral-7B. In the experiment, 7 benign clients possess the LMSYS-Chat dataset while 3 malicious clients possess the Beavertails dataset.

| Model | Llama2-7B | | | Mistral-7B | | |
|---|---|---|---|---|---|---|
| Metric ↑ | Rule | RM | MT-1 | Rule | RM | MT-1 |
| FedAvg (No Attack) | 82.88 | -1.72 | 4.19 | 89.81 | -1.08 | 5.26 |
| FedAvg | 49.81 | -2.97 | 4.14 | 45.77 | -2.97 | 5.15 |
| Krum | 55.38 | -2.88 | 4.16 | 54.04 | -2.76 | 5.24 |
| DnC | 55.96 | -2.90 | 4.00 | 51.15 | -2.75 | 5.29 |
| Ours: Level 2 | 77.31 | -0.99 | 4.23 | 89.04 | -0.90 | 5.04 |

