# OpenReview forum: "Emerging Safety Attack and Defense in Federated Instruction Tuning of Large Language Models"
_ICLR.cc/2025/Conference — ICLR 2025 Poster_

### Official Review · Reviewer_F3Ge · 2024-10-27

**Soundness:** 2
**Presentation:** 2
**Contribution:** 1
**Rating:** 3
**Confidence:** 4

**Summary:**

This manuscript  presents a safety attack against federated instruction tuning (FedIT) along with a corresponding defense method. The focus is on data-based attacks, where the attacker (one of the parties) can manipulate the training data while adhering to the training protocol, without directly modifying the model parameters. The attack is performed using simple instruction tuning with a malicious dataset (obtained from either public or generated data). To defend against this attack, the authors propose fine-tuning with aligned data after training. The experiments show the propose method outperforms some baselines.

**Strengths:**

*S1.* This manuscript identifies vulnerabilities in FedIT caused by malicious data, which could compromise the alignment of large language models (LLMs).

**Weaknesses:**

*W1.* The attack model requires further clarification regarding the feasibility and necessity of data-based attacks. Typically, in FedIT, the attacker is assumed to be one of the parties who can control both the parameters transferred to the server and the training data. However, in this manuscript, the attacker is assumed to manipulate only the training data while leaving the local model parameters untouched. Additional explanation is needed on how this assumption aligns with real-world applications.

*W2.* The related work section lacks thorough comparisons. The manuscript claims to be the first work on data-based attacks, yet it is unclear when a data-based approach is preferable to a model-based one (see W1). Therefore, other attacks (both model-based and data-based) should be discussed and compared in the experiments.

*W3.* The experiments based on OpenFedLLM are not convincing. It is recommended to use more state-of-the-art or widely adopted LLMs for experimentation. Furthermore, stronger baselines, such as attacks that directly tune the parameters (see W1), should be included.

*W4.* There is limited technical challenge addressed in this manuscript. Simple fine-tuning with malicious or aligned data is applied without any specific techniques or novel approaches, providing limited insights for future research.

**Questions:**

N/A

---

> ### Comment · Reviewer_F3Ge · 2024-11-23
>
> I appreciate the authors' response and the additional experiments, which address some of my concerns regarding W3.
>
> However, for W1 and W2, I remain skeptical about the claim that "data-based attacks are more stealthy than model-based attacks." If this claim were accurate, there should not be an assumption that an attacker can only control the data but cannot directly modify the model parameter, because modifying the parameters is likely easier to detect. Additionally, is there any citation supporting this claim? If this finding is novel to the paper, it should be supported by comprehensive experiments or analysis.
>
> Moreover, Table R1 requires further clarification, including:
>
> * Whether the model-based method is an existing published approach and the state-of-the-art model-based attack.
>
> * Discussion of the relevant literature on model-based attacks.
>
> * An explanation as to why model-based attacks have lower safety scores in the context of FedAvg (first row).

---

> ### Comment · Reviewer_F3Ge · 2024-11-25
>
> Thank you to the authors for their detailed response. However, two core claims still require further support:
>
> 1. _"Manipulating only the training data is more stealthy than manipulating model parameters."_
> 2. _"Data-based attacks can make the attacked LLMs still capable of generating fluent natural language, whereas model-based attacks cannot."_
>
> These claims are key motivations for the proposed data-based attack, and additional evidence or discussion is necessary to strengthen the argument. Specifically:
>
> For the first claim, more comprehensive experiments or citations are needed to support this strong statement, which suggests the superiority of data-based attacks over **all** model-based attacks. The rebuttal provided only one preliminary experiment, which is insufficient to back such a strong claim. To substantiate this, experiments should demonstrate that the proposed method consistently outperforms **all** model-based attacks or, at the very least, the **state-of-the-art** model-based attacks. While some references on model-based attacks were provided, none of these works explicitly conclude that data-based attacks are superior to model-based attacks.
>
> For the second claim, additional experiments are also needed to support this point. One of the primary metrics in evaluating backdoor attacks is the accuracy on the original task. While I understand that data-based attacks can ensure fluent language generation, I am not convinced that all model-based attacks **always** fail in this regard. This claim should be substantiated through comprehensive experiments comparing cutting-edge model-based attack methods.
>
> My concerns on the superiority of data-based attacks is because **data-based attacks can be considered a subset of model-based attacks**. Manipulating data is essentially a specific way of manipulating model updates, where model parameters are influenced through the gradients of malicious data. Therefore, I question whether data-based attacks inherently have an advantage.
>
> Moreover, my original concern (W4) regarding technical depth still stands. In the rebuttal, the technical challenge discussed by the authors lies in the presence of normal data similar to malicious unaligned data. However, this challenge is **less related to the federated learning** setting - it is also relevant to any data-poisoning attack on the alignment of centralized LLMs. If the proposed method can overcome this challenge, how does it compare with other data-poisoning attack methods in centralized fine-tuning? Additionally, can other centralized data-poisoning methods be adapted to the FL setting to address this challenge?
>
> I believe addressing these questions would significantly improve this manuscript.

---

> ### Comment · Reviewer_F3Ge · 2024-11-29
>
> I would like to thank the authors for their thoughtful response. During the discussion, I have come to understand that the data-based attack presented in the paper demonstrates certain advantages over other model-based attacks in classification tasks. However, my original concerns, specifically *W1* and *W2*, remain focused on the assumption outlined in the threat model that "the attacker can train its local model on an arbitrary training dataset" (line 199). To address this, I suggest the authors:
> 1. Remove this assumption from the threat model, and
> 2. Provide rigorous evidence of the proposed attack's advantage over other attacks in the revision.
>
> Regarding *W4*, I would like to clarify that I fully acknowledge the novelty of conducting data poisoning attacks in the context of FL. I have never questioned the novelty of this paper in my review. However, novelty alone is not the sole criterion for evaluating the quality of research. My concern lies with the **technical depth** or **contribution** of the study - namely, what new knowledge this paper introduces to the research community.
>
> The main insights I gained from this paper are as follows:
> 1. Data poisoning on clients' models can affect the global model through the federated averaging process (FedIT).
> 2. Fine-tuning a poisoned model with normal data can restore its normal behavior.
>
> It is widely recognized in the field that fine-tuning large language models (LLMs) with poisoned data can result in immoral or malicious outputs. Furthermore, many alignment strategies, including the simple approach of fine-tuning with normal data (as discussed in this paper), have already been extensively explored.
>
> Consequently, the insights presented in this paper offers limited contribution. The first insight - that poisoning a portion of a client’s data can influence the global model through federated learning - is not surprising. The second insight, which pertains to the general recovery of a poisoned model through fine-tuning with normal data, is unrelated to federated learning and reflects a known fact within the domain of LLM alignment.
>
> Therefore, while I recognize the novelty of this paper, I find its **contribution** and **technical depth** to be marginal and insufficient for publication in a top-tier conference. To strengthen the work, I recommend the authors explore how the attacks in FL differ from well-studied data poisoning attacks in centralized settings, for instance, focusing on how data heterogeneity in FL impacts these attacks.
>
> After careful consideration of the authors' response, I have decided to increase the rating on soundness but maintain my original overall rating.

---

### Official Review · Reviewer_J4BS · 2024-10-30

**Soundness:** 3
**Presentation:** 3
**Contribution:** 3
**Rating:** 6
**Confidence:** 2

**Summary:**

This paper reveals that the current form of FedIT presents vulnerability when malicious clients train their local models with harmful datasets, resulting in compromised safety alignment of LLMs. While existing defense methods fail to address such attacks, this paper proposes a post-hoc defense by fine-tuning central LLM with harmless instructions and responses that are generated by external LLMs without any human efforts. The evaluation demonstrates the effectiveness of the proposed attack and improves the safety alignment of the proposed defense.

**Strengths:**

1. The paper explores the safety topic in federated learning, which is important to understand especially in the era of LLMs.

2. The paper is well-written and the logical flow of the paper is clear and easy to follow.

3. The paper considers the scalability of the proposed work.

**Weaknesses:**

1. The authors should explain the assumption that malicious users’ objective is to make LLMs behave harmfully while given harmful instructions and behave normally when given normal instructions.

2. The proposed attack is clear, yet the experiment setup (i.e., 30% are malicious users) is missing explanations.

3. The tradeoffs and benefits of the proposed defense method over existing defense methods need to be clarified. A clear comparison would have been helpful.

**Questions:**

Thank the authors for their work. Please see my questions as follows.

1. Could the authors clarify the rationale behind the attacker's objective of making LLMs behave harmfully only for harmful instructions? Have they considered scenarios where attackers aim to make LLMs behave harmfully for all types of instructions?

2. The authors assumed that 30% of users are malicious. The authors also stated for each round, only 3 out of 10 users were sampled. Is it randomly sampled? Moreover, for some rounds, users are all normal or malicious, which means this sampling process will naturally deal with malicious attacks. Could the authors provide a sensitivity analysis on the percentage of malicious users? Additionally, it would be valuable to see a breakdown of performance for rounds with different compositions of benign and malicious users to better understand the impact of the sampling process on the attack and defense effectiveness.



3. The authors performed most experiments on LLAMA-7B, and it is good to compare with other models. Could the authors provide hypotheses or analyses on why different models show varying levels of vulnerability to the proposed attack, as seen in Fig. 4? This could help in understanding if certain model architectures or training approaches are inherently more robust to such attacks.

---

### Official Review · Reviewer_vfzj · 2024-11-04

**Soundness:** 3
**Presentation:** 3
**Contribution:** 3
**Rating:** 6
**Confidence:** 4

**Summary:**

This paper addresses attacks and defenses of safety alignment in federated instruction tuning. The authors propose a post-hoc defense method that fine-tunes the aggregated model with safe data at the server, thus restoring safety alignment without detecting malicious client models. Experiments demonstrate that the safety attack can significantly reduce the model’s safety rate, while the defense approach effectively restores the model's safety alignment.

**Strengths:**

1. This paper addresses safety alignment in federated instruction tuning, a problem that is both important and interesting. The authors propose attacks along with corresponding defenses to against the attacks.

2. The paper introduces a post-hoc defense mechanism that uses generated safe data to finetune the aggregated model and restore safety alignment without inspecting individual client models.

3. The authors conducted extensive experiments across multiple datasets and defense scenarios. The results show that the proposed attack can significantly compromise the model's alignment, while the defense method can restore alignment effectively.

**Weaknesses:**

1. The baseline defense methods used for comparison may not be well-suited to the attack in this paper. Methods like Krum, Median, and Trimmed Mean are designed primarily for Byzantine attacks that manipulate model weights, rather than poisoning local training datasets. Also, methods such as Foolsgold can be effective against backdoor attacks that might poison training data, e.g., label flipping attacks, but these methods were developed for classification tasks. However, this paper addresses text-generation tasks with different goals and attack mechanisms. Thus, I'm concerned about the relevance and effectiveness of these baseline methods.

2. The paper mentioned using a single NVIDIA RTX 3090 GPU to train models across 10 clients, with 3 clients participating training in each round. However, it's not clear how a single RTX 3090 can support fine-tuning three LLaMA2-7B simultaneously. Also, are the benign local models are trained with IID data?

3. The motivation for training LLMs with FL is unclear. While FL is primarily used to protect the privacy of local data, LLMs are often trained on publicly available datasets, which may reduce the need for privacy-preserving approaches in this context. I understand there may be sensitive scenarios, such as in medicine, law, or finance, where each FL participant wants to protect their data. However, these cases would involve different data distributions across FL clients, which are different from the experiment setting.

**Questions:**

Typically, applying defense mechanisms would lead to a reduction in accuracy. I'm curious if this also holds true for LMs. Since the approach involves using a different dataset for fine-tuning the aggregated model at the server, will this fine-tuning compromise the performance of the benign local models?

---

### Official Review · Reviewer_pZyk · 2024-11-04

**Soundness:** 2
**Presentation:** 2
**Contribution:** 3
**Rating:** 6
**Confidence:** 3

**Summary:**

The paper addresses the vulnerabilities in the safety alignment of large language models (LLMs) trained through federated instruction tuning (FedIT). It introduces a novel safety attack method that allows malicious clients to compromise the safety of LLMs by using unaligned datasets, which can be generated automatically without human intervention. Meanwhile, this paper proposes a post-hoc defense method that involves the server generating aligned data to fine-tune the LLMs, thereby improving their safety alignment. Extensive experimental results demonstrate the effectiveness of the proposed attack and defense methods.

**Strengths:**

1. This paper for the first time reveals the vulnerability of safety alignment of LLMs trained via federated instruction tuning.

2. This paper proposes the first data poisoning method that aims to compromise the safety of FedIT.

3. This paper proposes a post-hoc defense method that effectively enhances the safety alignment of compromised LLMs.

**Weaknesses:**

1. The baseline defense methods considered in this paper are all relatively outdated. It is better to consider the effect of the proposed attack under the defense method after 2022, such as [1][2].

2. This paper does not discuss whether the post-hoc defense will cause the performance degradation of the LLM model.

Reference

[1] Lu, Zhi et al. Split Aggregation: Lightweight Privacy-Preserving Federated Learning Resistant to Byzantine Attacks. TIFS’2024

[2] Wan, Wei et al. A Four-Pronged Defense Against Byzantine Attacks in Federated Learning. MM’2023

**Questions:**

1. In the experimental setups, the paper claims that four benign datasets were considered. However, I have not found the experimental results of the Dromedary-verbose and Wizard-evol datasets.

2. This paper proposes a post-hoc defense to remedy the damage caused by attacks during FL. I wonder if existing centralized fine-tuning defenses [3] can mitigate the proposed attack? What is the innovation of this paper in the defense method?

3. I want to know the performance of the proposed attack on different proportions of malicious clients.

Reference

[3] Liu, Kang et al. Fine-pruning: Defending against backdooring attacks on deep neural networks. RAID’2018

---

### Meta-Review · Area_Chair_nS7Z · 2024-12-20

**Metareview:**

The paper addresses the vulnerabilities in the safety alignment of large language models (LLMs) trained through federated instruction tuning (FedIT). It introduces a novel safety attack method that allows malicious clients to compromise the safety of LLMs by using unaligned datasets, which can be generated automatically without human intervention.

+ The paper addresses an important problem.
+ The paper is well-written.
+ The experiments are extensive.

- Some of the baselines are not suitable.
- Both the attack and the mitigation strategy appear not to be specific to federated learning.

**Additional Comments On Reviewer Discussion:**

The author rebuttal was carefully considered by the reviewers. It offers sufficient explanation for some of the technical details raised in the initial reviews. However, there is still a lack of clarity about the specific challenges offered by federated learning.

---

### Decision · Program_Chairs · 2025-01-22

Accept (Poster)